# Backpropagating Linearly Improves Transferability of Adversarial Examples

**Yiwen Guo** *
ByteDance AI Lab
guoyiwen.ai@bytedance.com

**Qizhang Li** *
ByteDance AI Lab
liqizhang@bytedance.com

**Hao Chen**
University of California, Davis
chen@ucdavis.edu

## Abstract

The vulnerability of deep neural networks (DNNs) to adversarial examples has drawn great attention from the community. In this paper, we study the transferability of such examples, which lays the foundation of many black-box attacks on DNNs. We revisit a not so new but definitely noteworthy hypothesis of Goodfellow *et al.*'s and disclose that the transferability can be enhanced by improving the linearity of DNNs in an appropriate manner. We introduce linear backpropagation (LinBP), a method that performs backpropagation in a more linear fashion using off-the-shelf attacks that exploit gradients. More specifically, it calculates forward as normal but backpropagates loss as if some nonlinear activations are not encountered in the forward pass. Experimental results demonstrate that this simple yet effective method obviously outperforms current state-of-the-arts in crafting transferable adversarial examples on CIFAR-10 and ImageNet, leading to more effective attacks on a variety of DNNs. Code at: `https://github.com/qizhangli/linbp-attack`.

## 1 Introduction

An adversarial example crafted by adding imperceptible perturbations to a natural image is capable of fooling the state-of-the-art models to make arbitrary predictions, indicating the vulnerability of deep models. The undesirable phenomenon not only causes a great deal of concern when deploying DNN models in security-sensitive applications (e.g., autonomous driving), but also sparks wide discussions about its theoretical understanding.

Some intriguing properties of the adversarial example has also been highlighted [46, 9], one of which that is of our particular interest is its transferability (across models), referring to the phenomenon that an adversarial example generated on one DNN model may fool many other models as well, even if their architectures are different. It acts as the core of transfer-based black-box attacks [37, 38, 31, 29, 35], and can also be utilized to enhance query-based attacks [7, 53, 22]. We attempt to provide more insights in understanding the black-box transfer of adversarial examples.

In this paper, we revisit the hypothesis of Goodfellow *et al.*'s [13] that the transferability (or say generalization ability) of adversarial examples comes from the "linear nature" of modern DNNs. We conduct empirical study to try utilizing the hypothesis for improving the transferability in practice. We identify a non-trivial improvement by simply removing some of the nonlinear activations in a DNN, shedding more light on the relationship between network architecture and adversarial threat. We also disclose that a more linear backpropagation solely suffices in improving the transferability. Based on the analyses, we introduce LinBP, a simple yet marvelously effective method that outperforms current state-of-the-arts in attacking a variety of victim models on CIFAR-10 and ImageNet, using different sorts of source models (including VGG-19 [44], ResNet-50 [18], and Inception v3 [45]).

## 2  Background and Related Work

Under different threat models, there exist white-box attacks and black-box attacks, where an adversary has different levels of access to the model information [38].

**White-box attacks.**  Given full access to the architecture and parameters of a victim model, white-box attacks are typically performed by leveraging gradient with respect to its inputs. Many methods aim to maximize the prediction loss $L(\mathbf{x} + \mathbf{r}, y)$ with a constraint on the $\ell_p$ norm of the perturbation, *i.e.*,

$$\max_{\|\mathbf{r}\|_p \leq \epsilon} L(\mathbf{x} + \mathbf{r}, y) \tag{1}$$

In a locally linear view, FGSM [13] lets the perturbation be $\epsilon \cdot \text{sgn}(\nabla_{\mathbf{x}} L(\mathbf{x}, y))$ for $p = \infty$, given a benign example $\mathbf{x} \in \mathbb{R}^n$ coupled with its label $y$. While highly efficient, the method exploits only a coarse approximation to the loss landscape and can easily fail when a small $\epsilon$ value is required. Aiming at more powerful attacks, I-FGSM [28] and PGD [32] are further introduced to generate adversarial examples in an iterative manner. In comparison to FGSM that only takes a single step along the direction of gradient (sign), these methods deliver superior attack performance on both normally trained and specifically regularized models, confirming that the multi-step scheme is more effective in the white-box setting. In addition to these methods that stick with the same prediction loss as for standard training, there exist attacks that use the gradient of some different functions and attempt to solve a dual problem, *e.g.*, DeepFool [34] and C&W's attack [3]. Other popular attacks include the universal adversarial attack [33], elastic-net attack [5], momentum iterative attack [11], just to name a few.

**Black-box attacks.**   In contrast to the white-box setting in which input gradients can be readily calculated using backpropagation, adversarial attacks in a black-box setting are more strenuous. In general, no more information than the prediction confidences on possible classes will be leaked to the adversary. Existing attacks in this category can further be divided into *query-based* methods and *transfer-based* methods. Most query-based methods share a common pipeline consisting of iteratively estimating gradients utilizing zeroth-order optimizations and mounting attacks as if in the white-box setting [6, 23, 24, 48, 36, 14] [2], while transfer-based methods craft adversarial examples on some source models and anticipate them to fool the victim models (also known as the target models). Of the two lines of research work, the former usually yields higher success rates, yet many of the methods suffer from high query complexity. Recently, an intersection of the two lines is also explored, leading to more query-efficient attacks [7, 53, 22].

**Transferability.**   As the core of many black-box attacks, the transferability of adversarial examples has been studied since the work of Goodfellow *et al.*'s [13], where it is ascribed to the *linear nature* of modern DNNs. Liu *et al.* [31] perform an insightful study and propose probably the first method to enhance it in practice, by crafting adversarial examples on an ensemble of multiple source models. A somewhat surprising observation is that single-step attacks are sometimes more transferable than their multi-step counterparts [28], that are definitely more powerful in the white-box setting. Over the past one or two years, efforts have been devoted to improving the transferability and facilitating multi-step attacks under such circumstances. For instance, Zhou *et al.* [55] suggest that optimization on an intermediate level makes multi-step adversarial attacks more transferable. They propose to maximize disturbance on feature maps extracted at multiple intermediate layers of a source model using mostly low-frequency perturbations if possible. Similarly, Inkawhich *et al.* [25] and Huang *et al.* [21] also propose to mount attacks on intermediate feature maps. Their methods, named activation attack (AA) and intermediate level attack (ILA), respectively, target on a single hidden layer and use different distortion measures. In supplementary material, we will provide a possible re-interpretation of the intermediate features-based attacks under the hypothesis regarding linearity. Another method that yields promising results is the diverse inputs I-FGSM (DI$^2$-FGSM) [50], which introduces diverse input patterns for I-FGSM.

Very recently, Wu *et al.* [49] show that packing less gradient into the main stream in ResNets makes adversarial examples more transferable, which inspires the re-normalization in our LinBP. Yet, unlike their method that is only functional on DNNs with skip connections, our method works favorably on a much broader range of architectures, and as will be shown in Section 5.3, our method can be easily combined with theirs to achieve even better results.

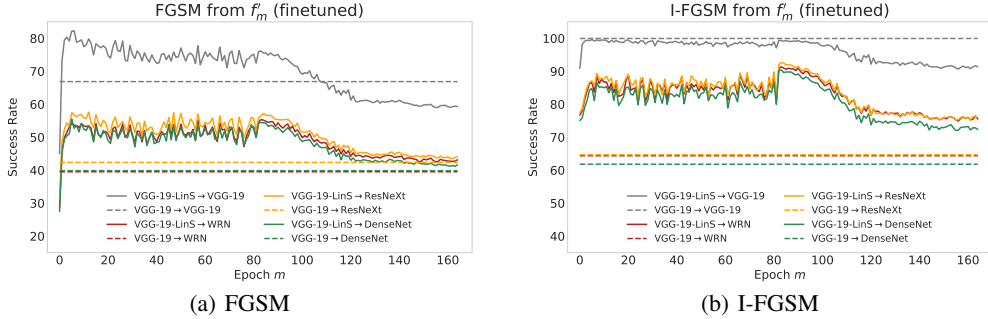

Figure 1: How the transferability of (a) FGSM and (b) I-FGSM examples changes with the number of fine-tuning epochs on $f'$. We perform $\ell_\infty$ attacks under $\epsilon = 0.03$. Best viewed in color.

## 3 The Linearity Hypothesis Revisited

In this section, we attempt to shed more light on the transferability of adversarial examples, which is still to be explored after all the prior study. Let us first revisit an early hypothesis made by Goodfellow *et al.* [13] that the cause of adversarial examples and their surprising transferability is the linear nature of modern DNNs, *i.e.*, they resemble linear models trained on the same dataset. Although the hypothesis seems plausible, there is of yet little empirical evidence on large networks that could verify it, let alone utilizing it in practice. We take a step towards providing more insights into it.

### 3.1 A More Linear Source Model

Given a source model $f : \mathbb{R}^n \to \mathbb{R}^c$ that is trained to classify instances from $c$ classes, a reasonable experiment is to compare the success rate of transfer-based attacks mounted on $f$ and a more linear model $f'$ (or a more non-linear model $f''$). For simplicity of notations, let us consider a source model parameterized by a series of weight matrices $W_1 \in \mathbb{R}^{n_0 \times n_1}, \ldots, W_d \in \mathbb{R}^{n_{d-1} \times n_d}$, in which $n_0 = n$ and $n_d = c$, whose output can be written as

$$f(\mathbf{x}) = W_d^T \sigma(W_{d-1}^T \ldots \sigma(W_1^T \mathbf{x})), \tag{2}$$

in which $\sigma$ is a non-linear activation function and it is often chosen as a rectified linear unit (ReLU) in modern DNNs. Such a representation of $f$ covers both multi-layer perceptrons and convolutional networks. Since its non-linearity solely comes from the $\sigma$ functions, apparently, we can get a desired $f'$ by simply removing some of them, leading to a model that shares the same number of parameters and core architecture with $f$. Note that unlike some work in which Taylor expansion is used and linearization is obtained locally [43, 15], our method (dubbed linear substitution, LinS) leads to a *global* approximation which is identical for different network inputs and more suitable to our setting.

The experiment is performed on CIFAR-10 [27] using VGG-19 [44] with batch normalization [26] as the original source model $f$. It shows an accuracy of $93.34\%$ on the benign test set. For victim models, we use a wide ResNet (WRN) [54], a ResNeXt [51], and a DenseNet [20]. Detailed experimental settings are deferred to Section 5.1. We remove all nonlinear units (*i.e.*, ReLUs) in the last two VGG blocks to produce an initial $f'$, denoted as $f'_0$. It can be written as the composition of two sub-nets, *i.e.*, $f'_0 = g'_0 \circ h$, in which $g'_0$ is purely linear.

Since such a straightforward "linearization" causes a deteriorating effect on the network accuracy in predicting benign instances, we try fine-tuning the LinS model $f'$. We evaluate the transferability of FGSM and I-FGSM examples crafted on $f'_0, \ldots, f'_m, \ldots$ obtained after $0, \ldots, m, \ldots$ epochs of **fine-tuning** and we summarize the results in Figure 1. It shows that our LinS method can indeed facilitate the transferability, especially with a *short* period of fine-tuning that substantially gains the prediction accuracy of the network, although somewhat surprisingly, a different trend emerges as the process further progresses. For $m \geq 1$, $f'_m$ always helps generate more transferable adversarial examples than $f$, and it is worthy of mentioning that I-FGSM examples crafted on $f'_0$ also achieve decent transferability. The most transferable (I-FGSM) examples are collected right after the learning rate is cut around $m = 80$. Further training leads to decreased transferability, on account of overfitting [40]. The converged model shows comparable adversarial transferability with that **trained from scratch**, *cf.* Figure 1 and 2. Without ReLU layers in the later blocks, both models produce more transferable examples, and hence the hypothesis is partially verified. We also provide possible interpretation of the intermediate-level attacks [25, 21] under the hypothesis in Section D in the supplementary material.

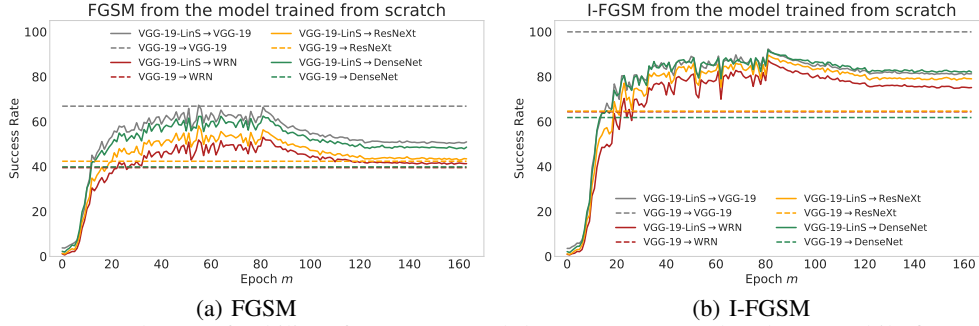

(a) FGSM                                          (b) I-FGSM

Figure 2: How the transferability of (a) FGSM and (b) I-FGSM examples changes while fine-tuning a more linear DNN from scratch. We perform $\ell_\infty$ attacks under $\epsilon = 0.03$. Best viewed in color.

We have shown experimental results of fine-tuning models and training models from scratch. Despite the similar final success rates on attacking WRN, ResNeXt, and DenseNet, their performance in attacking the VGG-19 source model (*i.e.*, $f$) is very different. More specifically, Figure 2 depicts that attacking the source model $f$ is as difficult as attacking the other victim models if training from scratch, indicating that fine-tuning endows $f'$ considerable ability to resemble $f$.

# 4   Our Linear Backpropagation Method

It has been confirmed that by directly removing ReLU layers, we can obtain improved transferability of adversarial attacks. Nevertheless, skipping more and more ReLUs does not always indicate better performance, since directly modifying the architecture as such inevitably degenerates the prediction accuracy. A more linear source model predicting poorly leads to inputs gradients unlike those of the victim models, and thus unsatisfactory transferability. We discuss how to seek a reasonable trade-off between the linearity and accuracy in Section 4.1 and take special care of model branching that are ubiquitous in advanced DNNs in Section 4.2.

## 4.1   Linear Backpropagation

First, we disentangle the effect of ReLU removal in computing forward and backward for crafting adversarial examples, and we study how the specific "linearization" affects both passes in a similar manner to prior work demystifying dropout [12]. Since it makes little sense to couple a "linearized" forward with a non-linear backpropagation, we focus on linear backpropagation (LinBP). Concretely, we calculate forward as normal but backpropagate the loss as if no ReLU is encountered in the forward pass, *i.e.*,

$$\tilde{\nabla}_{\mathbf{x}} L(\mathbf{x}, y) = \frac{dL(\mathbf{x}, y)}{d\mathbf{z}_g} W_d \dots W_k \frac{d\mathbf{z}_h}{d\mathbf{x}} \tag{3}$$

in which $\mathbf{z}_h := h(\mathbf{x})$, $g$ is the sub-net of $f$ being comprised of the $k$-th to $d$-th parameterized layers followed by ReLUs and $\mathbf{z}_g := g(\mathbf{z}_h) = W_d^T \sigma(W_{d-1}^T \dots \sigma(W_k^T \mathbf{z}_h) = f(\mathbf{x})$. It can be regarded as feeding more gradient into some linear path than the nonlinear path that filters out the gradient of the negative input entries. Such a LinBP method requires no fine-tuning since it calculates forward and makes predictions just like a well-trained source model $f$. Denote by $L'_m(\mathbf{x}, y)$ the prediction loss used by our LinS models as described in Section 3.1, *i.e.*, $L'_m = l \circ s \circ f'_m$. We compare the transferability using (a) $\nabla L'_0$, *i.e.*, input gradient calculated with $f'_0$, (b) $\nabla L'_{\text{opt}}$, *i.e.*, input gradient calculated with the optimal fine-tuning epoch, and (c) LinBP in Table 1. It can be seen that LinBP performs favorably well in comparison with LinS models with or without fine-tuning, achieving a more reasonable trade-off between the linearity and accuracy. They all show higher computational efficiency than the baseline due to the (partially) absence of ReLUs.

## 4.2   Gradient Branching with Care

In the previous sections of this paper, we discuss and empirically evaluate source models without auxiliary branches, *i.e.*, VGG-19. However, DNNs with skip-connections and multiple branches (as introduced in Inception [45] and ResNets [18]) have become ubiquitous in modern machine learning applications, for which it might be slightly different when performing LinBP. Specifically,

Table 1: Transferability of adversarial examples crafted originally using $\nabla L$ and in a more linear spirit using LinS and LinBP, utilizing single-step and multi-step attacks under $\epsilon = 0.03$. Note that for $\nabla L'_{\text{opt}}$, we test at the optimal fine-tuning epochs, which is unpractical and just for reference.

| Method | Gradient | VGG-19* (2015) | WRN (2016) | ResNeXt (2017) | DenseNet (2017) |
|---|---|---|---|---|---|
| FGSM | $\nabla L$ (original) | 66.68% | 39.52% | 42.36% | 39.86% |
| | $\nabla L'_0$ (LinS initial) | 45.00% | 28.70% | 30.96% | 27.42% |
| | $\nabla L'_{\text{opt}}$ (LinS optimal) | 82.28% | 55.38% | 57.60% | 55.50% |
| | $\nabla L$ (LinBP) | **92.50%** | **55.64%** | **59.70%** | **55.82%** |
| I-FGSM | $\nabla L$ (original) | 99.96% | 64.32% | 64.82% | 61.92% |
| | $\nabla L'_0$ (LinS initial) | 90.86% | 76.70% | 77.44% | 75.02% |
| | $\nabla L'_{\text{opt}}$ (LinS optimal) | 99.66% | 91.36% | **92.62%** | **90.44%** |
| | $\nabla L$ (LinBP) | **100.00%** | **91.60%** | 92.06% | 89.86% |

for a residual building block $\mathbf{z}_{i+1} = \mathbf{z}_i + W_{i+1}^T \sigma(W_i^T \mathbf{z}_i)$ as illustrated in Figure 3, the standard backpropagation calculates the derivative as $d\mathbf{z}_{i+1}/d\mathbf{z}_i = 1 + W_i M_i W_{i+1}$ whilst our "linearization" calculates $\Omega_i = 1 + W_i W_{i+1}$, in which the missing $M_i$ is a diagonal matrix whose entries are 1 if the corresponding entries of $W_i^T \mathbf{z}_i$ are positive and 0 otherwise [16, 15]. In later layers with a relatively large index $i$, $M_i$ can be very sparse and removing it during backpropagation results in less gradient flow through the skip-connections, which is undesired according to a recent study [49]. We alleviate the problem by re-normalizing the gradient passing backward through the main stream of the residual units, i.e., calculating $1 + \alpha_i W_i W_{i+1}$ instead during backpropagation, in which $\alpha_i = \|d\mathbf{z}_{i+1}/d\mathbf{z}_i - 1\|_2 / \|\Omega_i - 1\|_2$. The scalar $\alpha_i$ is thus layer-specific and automatically determined by gradients. In particular, for a ReLU layer whose input is a matrix/tensor without any negative entry, we have $M_i = I$ and $\alpha = 1$, which means that the backward pass in our LinBP will be the same as a standard backpropagation. DNNs with more branches are tackled in a similar spirit. Our re-normalization works in a different way from Wu *et al.*'s skip gradient method (SGM) [49], and we will demonstrate in Section 5.3 that combining them leads to further performance improvement.

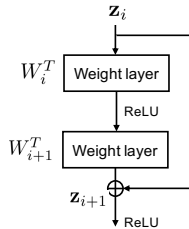

Figure 3: An example of the residual units in the ResNet family. Batch normalization are omitted in the figure for clarity reasons.

## 5 Experimental Results

In this section, we compare the proposed method with existing state-of-the-arts. We mainly compare with TAP [55] and ILA [21] that use intermediate level disturbance and SGM [49] that is specifically designed for source models equipped with skip connections (which can actually all be combined with our method). Table 2 and Table 3 summarize the comparison results using I-FGSM as the back-end attack, and FGSM results are reported in Table 6 and 7 in our supplementary material. Due to the space limit of the paper, results on stronger back-end attacks (including DI$^2$-FGSM [50], PGD [32], and an ensemble attack [31]) are also deferred to the supplementary material. It can be seen that our method outperforms competitors significantly in most test cases.

### 5.1 Experimental Settings

We focus on untargeted $\ell_\infty$ attacks on deep image classifiers. Different methods are compared on CIFAR-10 [27] and ImageNet [41], on the basis of single-step and multi-step attacks. On both datasets, we set the maximum perturbation as $\epsilon = 0.1, 0.05, 0.03$ to keep inline with ILA. We further provide results with $\epsilon = 16/255, 8/255, 4/255$ in Section F in the supplementary materials for comparison with contemporary methods tested in such a setting. We choose VGG-19 [44] with batch normalization and ResNet-50 [18] as source models for CIFAR-10 and ImageNet, respectively. For victim models

Table 2: Success rates of transfer-based attacks on *CIFAR-10* using I-FGSM with $\ell_\infty$ constraint in the untargeted setting. PyramidNet$^\dagger$ indicates PyramidNet [17]+ShakeDrop [52]+AutoAugment [8]. The source model is a VGG-19 with batch normalization and the symbol * indicates that the victim model is the same as the source model. Average is obtained on models different from the source.

| Dataset | Method | $\epsilon$ | VGG-19* (2015) | WRN (2016) | ResNeXt (2017) | DenseNet (2017) | PyramidNet$^\dagger$ (2019) | GDAS (2019) | Average |
|---------|--------|------------|----------------|------------|----------------|-----------------|------------------------------|-------------|---------|
| CIFAR-10 | I-FGSM | 0.1 | **100.00%** | 91.34% | 91.46% | 90.26% | 65.22% | 84.44% | 84.54% |
| | | 0.05 | 99.98% | 83.12% | 83.46% | 81.30% | 37.40% | 70.92% | 71.24% |
| | | 0.03 | 99.96% | 64.32% | 64.82% | 61.92% | 16.60% | 49.98% | 51.53% |
| | TAP+I-FGSM | 0.1 | **100.00%** | 94.68% | 95.98% | 92.70% | 91.00% | 89.76% | 92.82% |
| | | 0.05 | 99.76% | 94.84% | 94.54% | 94.32% | 69.70% | 87.78% | 88.24% |
| | | 0.03 | 98.16% | 85.06% | 87.86% | 84.60% | 40.00% | 72.14% | 73.93% |
| | ILA+I-FGSM | 0.1 | **100.00%** | 99.96% | 99.96% | 99.90% | 95.72% | 97.58% | 98.62% |
| | | 0.05 | 99.98% | 98.44% | 98.50% | 97.92% | 66.16% | 90.36% | 90.28% |
| | | 0.03 | 99.96% | 87.66% | 88.08% | 85.70% | 33.74% | 72.52% | 73.54% |
| | LinBP+I-FGSM | 0.1 | **100.00%** | **100.00%** | **100.00%** | **100.00%** | **97.06%** | **98.48%** | **99.11%** |
| | | 0.05 | **100.00%** | **99.18%** | **99.22%** | **98.92%** | **72.96%** | **93.86%** | **92.82%** |
| | | 0.03 | **100.00%** | **91.60%** | **92.06%** | **89.86%** | **41.30%** | **77.10%** | **78.38%** |

on CIFAR-10, we choose a 272-layer PyramidNet [17]+ShakeDrop [52]+AutoAugment [8] (denoted as PyramidNet$^\dagger$ in this paper), which is one of the most powerful classification models on CIFAR-10 trained using sophisticated data augmentation and regularization techniques, GDAS [10], a very recent model obtained from neural architecture search, and some widely applied DNN models including a WRN [54], a ResNeXt [51], and a DenseNet [20] (to be more specific, WRN-28-10, ResNeXt-29, and DenseNet-BC) $^3$. Note that all the victim models are invented after VGG-19 and their architectures are remarkably different, making the transfer of adversarial examples strikingly challenging in general. On ImageNet, we use Inception v3 [45], DenseNet [20], MobileNet v2 [42], SENet [19], and PNASNet [30] whose architecture is also found by neural architecture search. These models are directly collected from Torchvision [39] $^4$ and an open-source repository [53]. Their prediction accuracies on benign test sets are reported in the supplementary materials. Different victim models may require slightly different sizes of inputs, and we follow their official pre-processing pipeline while feeding the generated adversarial examples to them. For instance, for DenseNet on ImageNet, we first resize the images to $256 \times 256$ and then crop them to $224 \times 224$ at the center.

We follow prior arts and randomly sample 5000 test instances that could be classified correctly by all the victim models on each dataset and craft adversarial perturbations on them. For comparison, we evaluate the transfer rate of adversarial examples crafted using different methods to victim models based on the same single/multi-step attacks. Inputs to the source models are re-scaled to the numerical range of $[0.0, 1.0]$, and the intermediate results obtained at the end of each attack iteration are clipped to the range to ensure that valid images are given. When establishing a multi-step baseline using I-FGSM, the back-end attack that is invoked in Table 2 and 3, we run for 100 iterations on CIFAR-10 inputs and 300 iterations on ImageNet inputs with a step size of $1/255$ such that its performance reaches plateaus on both datasets. For competitors, we follow their official implementations and evaluate on the same 5000 randomly chosen test instances for fairness. The hidden layer where ILA is performed can be chosen on the source models following a guideline in [21], and our LinBP is invoked at the same position for fairness. Without further clarification, the last two convolutional blocks in VGG-19 and the last eight building blocks in ResNet-50 are modified to be more linear during backpropagation in our LinBP. On ResNet-50, we remove the first type of ReLUs in Figure 3 for simplicity and utilize re-normalization as introduced in Section 4.2. We let TAP [55], SGM [49], and our LinBP take the same number of running iterations as for the standard I-FGSM. A 100-iteration ILA [21] acting on the I-FGSM examples is compared, which is found to be superior to its default 10-iteration version. According to our results, no obvious performance gain can be obtained by further increasing the number of ILA running iterations. For the decay parameter in SGM, we choose it from $\{0.1, 0.2, 0.3, 0.4, 0.5, 0.6, 0.7, 0.8, 0.9\}$ and show in Table 3 the results with an optimal choice. All experiments are performed on an NVIDIA V100 GPU with code implemented using PyTorch [39].

## 5.2 Comparison to State-of-the-arts

Table 2 and 3 demonstrate that our method outperforms all the competitors in the context of average score on attacking 10 different victim models on CIFAR-10 and ImageNet using an I-FGSM back-end,

---

$^3$https://github.com/bearpaw/pytorch-classification
$^4$https://pytorch.org/docs/stable/torchvision/models.html

Table 3: Success rates of transfer-based attacks on *ImageNet* using I-FGSM with $\ell_\infty$ constraint under the untargeted setting. The source model is a ResNet-50 and the symbol * indicates that the victim model is the same as the source model. Average is obtained from models different from the source.

| Dataset | Method | $\epsilon$ | ResNet* (2016) | Inception v3 (2016) | DenseNet (2017) | MobileNet v2 (2018) | PNASNet (2018) | SENet (2018) | Average |
|---|---|---|---|---|---|---|---|---|---|
| ImageNet | I-FGSM | 0.1 | **100.00%** | 51.18% | 74.66% | 77.44% | 50.44% | 61.66% | 63.08% |
| | | 0.05 | **100.00%** | 27.78% | 51.34% | 54.92% | 25.26% | 33.90% | 38.64% |
| | | 0.03 | **100.00%** | 14.48% | 31.78% | 35.56% | 12.28% | 16.80% | 22.18% |
| | TAP+I-FGSM | 0.1 | **100.00%** | 89.28% | 97.82% | 98.10% | 89.72% | 95.54% | 94.09% |
| | | 0.05 | **100.00%** | 50.50% | 75.72% | 80.58% | 48.68% | 65.14% | 64.12% |
| | | 0.03 | **100.00%** | 22.86% | 44.34% | 52.22% | 19.58% | 29.88% | 33.78% |
| | ILA+I-FGSM | 0.1 | **100.00%** | 86.28% | 97.14% | 97.48% | 89.62% | 95.28% | 93.16% |
| | | 0.05 | **100.00%** | 51.62% | 79.66% | 82.20% | 56.14% | 69.18% | 67.76% |
| | | 0.03 | **100.00%** | 26.20% | 53.98% | 57.74% | 28.46% | 38.52% | 40.98% |
| | SGM+I-FGSM | 0.1 | **100.00%** | 68.30% | 89.32% | 93.64% | 72.02% | 83.74% | 81.40% |
| | | 0.05 | **100.00%** | 35.26% | 62.96% | 74.14% | 37.58% | 50.64% | 52.12% |
| | | 0.03 | **100.00%** | 19.20% | 41.06% | 52.66% | 19.30% | 27.92% | 32.03% |
| | LinBP+I-FGSM | 0.1 | **100.00%** | **93.84%** | **99.26%** | **98.74%** | **95.14%** | **97.46%** | **96.89%** |
| | | 0.05 | **100.00%** | **61.02%** | **89.18%** | **88.70%** | **62.66%** | **76.08%** | **75.53%** |
| | | 0.03 | **100.00%** | **30.80%** | **63.50%** | **64.18%** | **29.96%** | **42.74%** | **46.24%** |

under all the concerned $\epsilon$ values on all test cases. Echo prior observations in the literature, our results demonstrate that PyramidNet[†] is the most robust model to adversarial examples (transferred from VGG-19) on CIFAR-10. It is also the deepest model trained using sophisticated data augmentation and regularization techniques. Yet, our method still achieves nearly 73% success rate when performing untargeted $\ell_\infty$ attacks under $\epsilon = 0.05$. On ImageNet, Inception v3 and PNASNet seem to be the most robust models to ResNet-50 adversarial examples, even without adversarial training [13, 32, 47]. Our method achieves a success rate of 61.02% and 62.66% on them under $\epsilon = 0.05$. Comparison results using the FGSM back-end are given in our supplementary material. It can also be seen that our LinBP sometimes leads to even higher success rates than that obtained in the white-box setting (see the column of VGG-19* in Table 2). This was observed in the ILA experiments similarly [21], and can be explained if LinBP is viewed as an instantiation of backward pass differentiable approximation [1]. Other source models are also tested, and the same conclusions can be drawn (*e.g.*, from an Inception v3 source model, under $\epsilon = 0.05$, LinBP: 71.10% and ILA: 66.96% on average).

We further compare different methods in attacking an Inception model guarded by ensemble adversarial training [47] (https://bit.ly/2XKfrkz), which is effective in resisting transfer-based attacks. The superiority of LinBP holds on different $\epsilon$ values with the ResNet-50 source model. For $\epsilon = 0.1, 0.05$, and 0.03, it obtains 76.06%, 31.32%, and 14.32%, respectively. The second best is ILA (61.46%, 25.56%, and 12.74%). We also attack a robust ResNet (https://github.com/MadryLab/robustness) guarded by PGD adversarial training [32], and similar superiority of LinBP (*victim error rate*: 48.60%, 39.92%, and 37.10% with $\epsilon$=0.1, 0.05, 0.03) to ILA (42.30%, 37.82%, and 36.60%) is obtained.

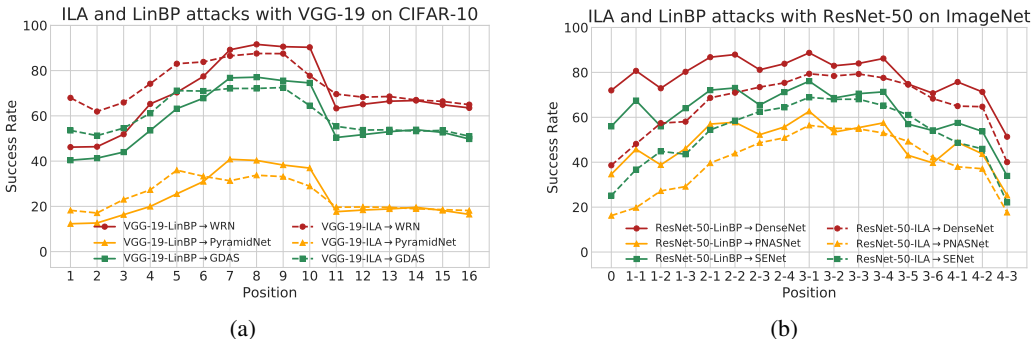

(a)                                                                                          (b)

Figure 4: How the performance of ILA and our LinBP varies with the choice of position on (a) CIFAR-10 trained VGG-19 and (b) ImageNet trained ResNet. "3-1" on ResNet-50 means the first residual unit in the third meta block.

Since the performance of both our method and ILA varies with the position from where the network is modified. For our method, it is $k$ in Eq. (3) indicating how the sub-nets $g$ and $h$ are split from $f$, and for ILA, it indicates where to evaluate the disturbance, recall that ILA ignores the following layers and we "linearize" them by removing nonlinear activations. Here we perform a more detailed comparison by testing at different DNN layers and plotting how the performance of both methods

Table 4: Success rates of *combined methods on ImageNet*. The source model is a ResNet-50 and the symbol * indicates that the victim model is the same as the source model.

| Dataset | Method | $\epsilon$ | ResNet* (2016) | Inception v3 (2016) | DenseNet (2017) | MobileNet v2 (2018) | PNASNet (2018) | SENet (2018) | Average |
|---|---|---|---|---|---|---|---|---|---|
| ImageNet | SGM+I-FGSM | 0.1 | **100.00%** | 68.30% | 89.32% | 93.64% | 72.02% | 83.74% | 81.40% |
| | | 0.05 | **100.00%** | 35.26% | 62.96% | 74.14% | 37.58% | 50.64% | 52.12% |
| | | 0.03 | **100.00%** | 19.20% | 41.06% | 52.66% | 19.30% | 27.92% | 32.03% |
| | LinBP+I-FGSM+SGM ($\lambda = 0.3$) | 0.1 | **100.00%** | 93.68% | 99.48% | 98.96% | 95.22% | 97.72% | 97.01% |
| | | 0.05 | **100.00%** | 61.56% | 90.48% | 89.30% | 67.02% | 77.52% | 77.18% |
| | | 0.03 | **100.00%** | 33.08% | 67.98% | 67.68% | 35.36% | 47.68% | 50.36% |
| | LinBP+I-FGSM+SGM ($\lambda = 0.5$) | 0.1 | **100.00%** | **95.10%** | **99.48%** | **99.04%** | **96.28%** | **97.92%** | **97.56%** |
| | | 0.05 | **100.00%** | **63.98%** | **92.16%** | **90.20%** | **68.64%** | **79.40%** | **78.88%** |
| | | 0.03 | **100.00%** | **35.16%** | **70.84%** | **70.72%** | **35.60%** | **47.86%** | **52.04%** |
| ImageNet | ILA+I-FGSM | 0.1 | **100.00%** | 86.28% | 97.14% | 97.48% | 89.62% | 95.28% | 93.16% |
| | | 0.05 | **100.00%** | 51.62% | 79.66% | 82.20% | 56.14% | 69.18% | 67.76% |
| | | 0.03 | **100.00%** | 26.20% | 53.98% | 57.74% | 28.46% | 38.52% | 40.98% |
| | LinBP+I-FGSM+ILA | 0.1 | **100.00%** | 97.06% | 99.58% | 99.44% | 97.86% | 98.42% | 98.47% |
| | | 0.05 | **100.00%** | 71.64% | 93.68% | 92.24% | 75.62% | 84.24% | 83.62% |
| | | 0.03 | **100.00%** | 37.54% | 71.12% | 71.34% | 39.38% | 51.46% | 54.17% |
| | LinBP+I-FGSM+ILA+SGM | 0.1 | **100.00%** | **97.26%** | **99.74%** | **99.50%** | **98.06%** | **98.76%** | **98.66%** |
| | | 0.05 | **100.00%** | **72.92%** | **94.70%** | **93.62%** | **77.54%** | **84.80%** | **84.72%** |
| | | 0.03 | **100.00%** | **40.42%** | **75.02%** | **74.90%** | **43.08%** | **54.78%** | **57.64%** |

varies with such a setting. We compare on three victim models each for CIFAR-10 and ImageNet and illustrate the results in Figure 4. VGG-19 and ResNet-50 are still chosen as the source models. It can be easily observed that our LinBP achieves similar or better results than ILA on preferred positions on the source models.

## 5.3 Combination with Existing Methods

It seems possible to combine our LinBP with some of the prior work, *e.g.*, SGM [49] and ILA [21] and facilitate the transferability of adversarial examples further, since our method retains the core architecture of the source models and has re-normalized the gradient flow. The combination with ILA can be harmonious by performing LinBP first and use its results as directional guides. See the lower half of Table 4 for detailed results. Apparently, such a combination works favorably well in different settings. The combination of LinBP and SGM is also straightforward, by directly introducing another scaling factor in the residual units to be modified. Take the concerned building block in Section 4.2 as an example, we calculate $1 + \lambda\alpha_i W_i W_{i+1}$ in which $0 < \lambda < 1$ is the decay parameter for SGM. We evaluate such a combination with $\lambda = 0.3$ and $0.5$ and report the results in the upper half of Table 4. It shows that our method benefits from the further incorporation of SGM.

**Stronger back-end attacks.** Following prior arts [55, 21], we have reported results using I-FGSM and FGSM as representative single-step and multi-step back-end attacks respectively when comparing with competitors in Table 2 and 3 in Section 5.2, and Table 6 and 7 in the supplementary material. We would like to emphasize that there exist other choices for back-end attacks, and some of them can be more powerful and suitable for transfer-based attacks, *e.g.*, DI$^2$-FGSM [50], which is specifically designed for the task by introducing input diversity and randomness. We test our LinBP in conjunction with DI$^2$-FGSM, PGD, and an ensemble attack [31] in the supplementary material and observe that the conclusions from our main experiments hold.

## 6 Conclusion

In this paper, we attempt to shed more light on the transferability of adversarial examples across DNN models. We revisit an early hypothesis made by Goodfellow *et al.* [13] that the higher than expected transferability of adversarial examples comes from the linear nature of DNNs and perform empirical study to try utilizing it. Inspired by the experimental findings, we propose LinBP, a method that is capable of improving the transferability, by calculating forward as normal yet backpropagate the loss linearly as if there is no ReLU encountered. This simple yet very effective method drastically enhances the transferability of FGSM, I-FGSM, DI$^2$-FGSM, and PGD adversarial examples on a variety of different victim models on CIFAR-10 and ImageNet. We also show that it is orthogonal to some of the compared methods, combining with which we can obtain nearly $85\%$ average success rate in attacking the ImageNet models, under $\epsilon = 0.05$.

## Broader Impact

This work can potentially contribute to deeper understanding of DNN models and the adversarial phenomenon. The reliability of different models are compared under practical adversarial attacks, making it possible for commercial machine-learning-as-a-service platforms to choose more suitable models for security-critical applications.

## Acknowledgment

We would like to thank the anonymous reviewers for valuable suggestions and comments. This material is based upon work supported by the National Science Foundation under Grant No. 1801751. This research was partially sponsored by the Combat Capabilities Development Command Army Research Laboratory and was accomplished under Cooperative Agreement Number W911NF-13-2-0045 (ARL Cyber Security CRA). The views and conclusions contained in this document are those of the authors and should not be interpreted as representing the official policies, either expressed or implied, of the Combat Capabilities Development Command Army Research Laboratory or the U.S. Government. The U.S. Government is authorized to reproduce and distribute reprints for Government purposes not withstanding any copyright notation here on.

## Footnotes

*The first two authors contributed equally to this work, and Yiwen Guo is the corresponding author.

[2]Except those designed for an extreme scenario where only the final label prediction is available to an adversary [2, 4].

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
