[Supplementary Material]

# Backpropagating Linearly Improves Transferability of Adversarial Examples (Supplementary Material)

**Yiwen Guo** [*]
ByteDance AI Lab
guoyiwen.ai@bytedance.com

**Qizhang Li** [*]
ByteDance AI Lab
liqizhang@bytedance.com

**Hao Chen**
University of California, Davis
chen@ucdavis.edu

## A  Fine-tuning $f'$ with Frozen $h$

Empirical results in Section 3.1 in the main paper show that simply removing ReLUs lead to improved transferability. In this section, we try freezing all learnable parameters in the unmodified sub-net $h$ during fine-tuning and a similar observation about the initial improvement of transferability can still be made (see Figure 5). Yet, we see obviously in Figure 5 that the curves bear a longer period of decrease and finally the obtained success rates are even worse than those of the baseline, owing to its lower learning capacity if $h$ is frozen. Classification loss of these modified VGG-19 models on the benign CIFAR-10 test set is also reported, in Figure 6.

(a) FGSM

(b) I-FGSM

Figure 5: How the transferability of (a) FGSM and (b) I-FGSM adversarial examples changes while fine-tuning $g'$ (*i.e.*, the purely linear blocks of $f'$). $h$ is frozen during the fine-tuning process. We perform untargeted attacks under the constraint of $\ell_\infty$ constraints and $\epsilon$ is set as 0.03.

Figure 6: Test classification loss of different source models tested here and in Section 3.1.

---

[*]The first two authors contributed equally to this work, and Yiwen Guo is the corresponding author.

## B Benign-test-set Performance of The Victim Models

Table 5: Benign-test-set accuracy of the models. On ImageNet, it is evaluated on the $50\,000$ official validation images.

| Dataset | VGG-19 (2015) | WRN (2016) | ResNeXt (2017) | DenseNet (2017) | PyramidNet[†] (2019) | GDAS (2019) |
|---|---|---|---|---|---|---|
| CIFAR-10 | 93.34% | 96.21% | 96.31% | 96.68% | 98.44% | 97.19% |

| Dataset | ResNet (2016) | Inception v3 (2016) | DenseNet (2017) | MobileNet v2 (2018) | PNASNet (2018) | SENet (2018) |
|---|---|---|---|---|---|---|
| ImageNet | 76.15% | 77.45% | 74.65% | 71.88% | 82.90% | 81.32% |

## C Re-interpret Intermediate Feature-based Attacks?

As mentioned in the main paper, many recent successes in improving adversarial transferability benefit from maximizing intermediate level distortions rather than the final prediction losses [8, 3, 2] of DNNs. Taking ILA [2] as an example, it opts to maximizing the projection of disturbance $h(\mathbf{x} + \mathbf{r}) - h(\mathbf{x})$ on a guidance direction $\mathbf{v} := h(\mathbf{x} + \mathbf{r}_0) - h(\mathbf{x})$, given a prior adversarial perturbation $\mathbf{r}_0$, *i.e.*,

$$\max_{\|\mathbf{r}\|_p \leq \epsilon} J(\mathbf{x} + \mathbf{r}, y) \quad \text{s.t.,} \quad J(\mathbf{x} + \mathbf{r}, y) = \mathbf{v}^T(h(\mathbf{x} + \mathbf{r}) - h(\mathbf{x})). \tag{4}$$

Recall that many adversarial attacks attempt to solve the optimization problem (1). With the common decomposed representation $f = g \circ h$, we can write $L = l \circ s \circ g \circ h$, in which $s$ is the softmax function and $l$ is the cross-entropy loss that takes both the output of $s$ and the one-hot encoding of $y$ as inputs. Apparently, $J$ in Eq. (4) is more linear than $L$ on account of the additional non-linearity of $l$, $s$, and $g$. It can be considered as substituting $s \circ g$ with an identity mapping and replacing $l$ with a linear loss. Likewise, AA [3] also exploits an identity mapping, in combination with a quadratic loss though. Whether their practical effectiveness partially comes from higher linearity worth exploring in future work.

## D Single-step Back-end Attacks

When a single-step back-end attack FGSM is adopted, different methods are compared in Table 6 and 7. It can be seen that the superiority of method holds in most cases as with I-FGSM. It is worthy of mentioning that ILA is performed after the FGSM examples are obtained in advance, and thus it takes one more step in comparison to LinBP and others. We verified this by using a two-step policy for our LinBP on CIFAR-10, and it showed that our LinBP indeed achieved higher average success rates (90.49%, 74.44%, and 52.95%) than those of ILA in Table 6 (85.72%, 70.94%, and 50.68%).

Table 6: Success rates of transfer-based attacks on *CIFAR-10* using FGSM with $\ell_\infty$ constraints in the untargeted setting. TAP boils down to FGSM in the single-step setting hence their performance are the same. For ILA, we test it on the basis of FGSM examples crafted offline and take one more step following its optimization objective, *i.e., two steps in total*. The source model is a VGG-19 with batch normalization and the symbol * indicates that the victim model is the same as the source model. Average is obtained from models different from the source.

| Dataset | Method | $\epsilon$ | VGG-19* (2015) | WRN (2016) | ResNeXt (2017) | DenseNet (2017) | PyramidNet[†] (2019) | GDAS (2019) | Average |
|---|---|---|---|---|---|---|---|---|---|
| CIFAR-10 | FGSM | 0.1 | 82.52% | 74.98% | 81.64% | 84.28% | 61.34% | 88.86% | 78.22% |
| | | 0.05 | 73.38% | 55.34% | 61.64% | 59.00% | 32.50% | 67.18% | 55.13% |
| | | 0.03 | 66.68% | 39.52% | 42.36% | 39.86% | 17.42% | 39.00% | 35.63% |
| | TAP+FGSM | 0.1 | 82.52% | 74.98% | 81.64% | 84.28% | 61.34% | 88.86% | 78.22% |
| | | 0.05 | 73.38% | 55.34% | 61.64% | 59.00% | 32.50% | 67.18% | 55.13% |
| | | 0.03 | 66.68% | 39.52% | 42.36% | 39.86% | 17.42% | 39.00% | 35.63% |
| | ILA+FGSM | 0.1 | 88.92% | **85.32%** | **87.46%** | **89.30%** | **76.68%** | **89.82%** | **85.72%** |
| | | 0.05 | 87.14% | 73.9% | 76.48% | **76.82%** | **48.12%** | **79.40%** | **70.94%** |
| | | 0.03 | 82.04% | **55.98%** | 59.64% | **58.36%** | **24.76%** | **54.66%** | **50.68%** |
| | LinBP+FGSM | 0.1 | **92.06%** | 84.16% | 87.24% | 87.76% | 72.36% | 89.38% | 84.18% |
| | | 0.05 | **95.18%** | **73.96%** | **77.60%** | 76.34% | 43.76% | 77.50% | 69.83% |
| | | 0.03 | **92.50%** | 55.64% | **59.70%** | 55.82% | 22.30% | 52.28% | 49.15% |

Table 7: Success rates of transfer-based attacks on *ImageNet* using FGSM with $\ell_\infty$ constraints. TAP boils down to FGSM in the single-step setting thus their performance are the same. For ILA, we test it on the basis of FGSM examples crafted in advance and take one more step following its optimization objective, *i.e., two steps in total*. The source model is a ResNet-50 and the symbol * indicates that the victim model is the same as the source. Average is obtained from models different from the source.

| Dataset | Method | $\epsilon$ | ResNet* (2016) | Inception v3 (2016) | DenseNet (2017) | MobileNet v2 (2018) | PNASNet (2018) | SENet (2018) | Average |
|---|---|---|---|---|---|---|---|---|---|
| ImageNet | FGSM | 0.1 | 88.64% | 47.88% | 63.86% | 79.80% | 39.30% | 48.96% | 55.96% |
| | | 0.05 | 86.46% | 32.52% | 44.96% | 50.34% | 24.36% | 28.26% | 36.09% |
| | | 0.03 | 84.38% | 23.04% | 35.26% | 37.72% | 16.44% | 18.74% | 26.24% |
| | TAP+FGSM | 0.1 | 88.64% | 47.88% | 63.86% | 79.80% | 39.30% | 48.96% | 55.96% |
| | | 0.05 | 86.46% | 32.52% | 44.96% | 50.34% | 24.36% | 28.26% | 36.09% |
| | | 0.03 | 84.38% | 23.04% | 35.26% | 37.72% | 16.44% | 18.74% | 26.24% |
| | ILA+FGSM | 0.1 | 84.38% | 50.40% | 67.10% | 82.12% | 39.10% | 51.60% | 58.86% |
| | | 0.05 | 74.56% | 30.56% | 44.14% | 55.44% | 19.98% | 27.28% | 35.48% |
| | | 0.03 | 74.38% | 20.68% | 32.64% | 38.66% | 13.26% | 16.54% | 24.36% |
| | SGM+FGSM | 0.1 | 85.00% | 44.66% | 61.04% | 78.84% | 35.26% | 46.22% | 53.20% |
| | | 0.05 | 84.30% | 28.04% | 40.90% | 50.74% | 19.88% | 25.68% | 33.05% |
| | | 0.03 | 82.96% | 19.72% | 31.74% | 37.72% | 12.70% | 16.42% | 23.66% |
| | LinBP+FGSM | 0.1 | **91.42%** | **52.88%** | **69.44%** | **83.98%** | **42.66%** | **52.72%** | **60.34%** |
| | | 0.05 | **90.52%** | **34.80%** | **49.24%** | **56.26%** | **25.78%** | **31.18%** | **39.45%** |
| | | 0.03 | **88.56%** | **24.80%** | **39.28%** | **41.68%** | **17.32%** | **19.94%** | **28.60%** |

## E  Stronger Multi-step Back-end Attacks

In this section, we test more powerful back-end attacks for our LinBP, including $DI^2$-FGSM (in which the momentum mechanism is also incorporated and thus it is more powerful than the momentum iterative FGSM [1] in the concerned setting) and PGD. Note that PGD tested here incorporated randomness at each of its optimization iterations, as such randomness is shown to be beneficial to the adversarial transferability in experiments. We observe that our LinBP works better in conjunction with them than with I-FGSM. See Table 8 and 9 for detailed results when combining our LinBP with $DI^2$-FGSM and PGD, respectively. We also consider an ensemble adversarial attack [6] on ImageNet, utilizing ResNet-50 and Inception v3 as source models, with the help of which we achieve an average success rate of 98.17%, 78.94%, and 49.42%, under $\epsilon = 0.1, 0.05$, and $0.03$, respectively.

Table 8: Success rates of transfer-based attacks on *ImageNet* using $DI^2$-FGSM. The source model is a ResNet-50 and the symbol * indicates that the victim model is the same as the source model. The mean and standard deviation results of five runs are reported. Average is obtained from models different from the source.

| Dataset | Method | $\epsilon$ | ResNet* (2016) | Inception v3 (2016) | DenseNet (2017) | MobileNet v2 (2018) | PNASNet (2018) | SENet (2018) | Average |
|---|---|---|---|---|---|---|---|---|---|
| ImageNet | $DI^2$-FGSM | 0.1 | **100.00%±0.00%** | 68.06%±0.35% | 89.56%±0.21% | 90.10%±0.16% | 69.76%±0.22% | 76.04%±0.12% | 78.70% |
| | | 0.05 | **100.00%±0.00%** | 39.60%±0.32% | 67.62%±0.18% | 67.32%±0.12% | 39.66%±0.28% | 46.88%±0.22% | 52.22% |
| | | 0.03 | **100.00%±0.00%** | 23.40%±0.26% | 48.02%±0.28% | 48.12%±0.26% | 22.28%±0.34% | 25.86%±0.22% | 33.54% |
| | LinBP+$DI^2$-FGSM | 0.1 | **100.00%±0.00%** | **96.04%±0.09%** | **99.58%±0.12%** | **99.12%±0.05%** | **96.34%±0.15%** | **97.52%±0.08%** | **97.72%** |
| | | 0.05 | **100.00%±0.00%** | **68.30%±0.24%** | **92.14%±0.13%** | **91.20%±0.18%** | **68.76%±0.16%** | **79.34%±0.23%** | **79.95%** |
| | | 0.03 | **100.00%±0.00%** | **37.98%±0.28%** | **71.48%±0.33%** | **71.06%±0.29%** | **36.16%±0.25%** | **47.74%±0.27%** | **52.88%** |

Table 9: Success rates of transfer-based attacks on *ImageNet* using PGD. The source model is a ResNet-50 and the symbol * indicates that the victim model is the same as the source model. The mean and standard deviation results of five runs are reported. Average is obtained from models different from the source.

| Dataset | Method | $\epsilon$ | ResNet* (2016) | Inception v3 (2016) | DenseNet (2017) | MobileNet v2 (2018) | PNASNet (2018) | SENet (2018) | Average |
|---|---|---|---|---|---|---|---|---|---|
| ImageNet | PGD | 0.1 | **100.00%±0.00%** | 68.34%±0.17% | 90.02%±0.29% | 87.72%±0.16% | 67.71%±0.46% | 73.32%±0.25% | 77.42% |
| | | 0.05 | **100.00%±0.00%** | 38.68%±0.30% | 66.46%±0.43% | 65.86%±0.4% | 35.40%±0.26% | 43.49%±0.36% | 49.98% |
| | | 0.03 | **100.00%±0.00%** | 20.80%±0.22% | 43.04%±0.37% | 44.88%±0.2% | 17.33%±0.33% | 22.86%±0.12% | 29.78% |
| | LinBP+PGD | 0.1 | **100.00%±0.00%** | **95.70%±0.23%** | **99.55%±0.07%** | **99.23%±0.05%** | **95.44%±0.13%** | **97.51%±0.09%** | **97.49%** |
| | | 0.05 | **100.00%±0.00%** | **72.72%±0.41%** | **93.70%±0.07%** | **93.05%±0.15%** | **70.07%±0.36%** | **81.79%±0.23%** | **82.27%** |
| | | 0.03 | **100.00%±0.00%** | **41.65%±0.37%** | **74.15%±0.20%** | **74.14%±0.26%** | **37.57%±0.48%** | **52.05%±0.31%** | **55.91%** |

## F  Other $\epsilon$ Settings and More Source Models

We notice that there exist different setting in evaluations of adversarial attacks in recent papers, some use $\epsilon = 0.1, 0.08, 0.05, 0.04$, and $0.03$ as in the main body of our paper [2, 5, 4], and some others use $\epsilon = 16/255, 8/255$, and $4/255$, et cetera [8, 7]. Here we also report the performance of our

method under $\epsilon = 16/255$, $8/255$, and $4/255$ for easier comparison with contemporary methods and other comparable methods. We also considered more source architectures for comparison, including ResNet-18 on CIFAR-10 and Inception v3 on ImageNet. The results are shown in Table 10, Table 11, Table 12, and Table 13, in which PGD is used as the baseline, since it is more powerful than I-FGSM, and other methods were performed on the basis of PGD, just like in Table 9. Here we give results in both the untargeted and the targeted settings. Results on the basis of other back-end attacks are similar, and the superiority of our LinBP holds in most test cases.

Table 10: Success rates of transfer-based attacks on *CIFAR-10* using PGD with $\ell_\infty$ constraints. The source model is a *VGG-19*. Average is obtained from models different from the source.

| Attack | Method | $\epsilon$ | VGG-19* (2015) | WRN (2016) | ResNeXt (2017) | DenseNet (2017) | PyramidNet[†] (2019) | GDAS (2019) | Average |
|---|---|---|---|---|---|---|---|---|---|
| Untargeted (VGG-19) | PGD | 16/255 | 99.98% | 95.78% | 94.86% | 93.44% | 60.64% | 87.96% | 86.54% |
| | | 8/255 | 99.92% | 74.94% | 75.22% | 71.44% | 22.66% | 60.66% | 60.98% |
| | | 4/255 | 99.42% | 36.20% | 37.16% | 34.40% | 6.36% | 25.38% | 27.90% |
| | ILA+PGD | 16/255 | **100.00%** | 99.46% | 99.64% | 99.18% | 80.54% | 94.92% | 94.75% |
| | | 8/255 | 99.96% | 90.06% | 91.04% | 88.44% | 36.84% | 75.44% | 76.36% |
| | | 4/255 | 99.20% | 53.18% | 53.88% | 50.22% | 10.12% | 36.34% | 40.75% |
| | LinBP+PGD | 16/255 | **100.00%** | **99.90%** | **99.94%** | **99.86%** | **91.54%** | **98.44%** | **97.94%** |
| | | 8/255 | **100.00%** | **95.42%** | **95.76%** | **94.16%** | **50.46%** | **84.50%** | **84.06%** |
| | | 4/255 | **99.80%** | **58.64%** | **61.28%** | **56.42%** | **12.94%** | **42.80%** | **46.42%** |
| Targeted (VGG-19) | PGD | 16/255 | 98.82% | 84.34% | 78.00% | 74.28% | 30.34% | 52.30% | 63.85% |
| | | 8/255 | 96.12% | 54.40% | 49.90% | 46.94% | 8.44% | 29.72% | 37.88% |
| | | 4/255 | 93.62% | 17.22% | 15.08% | 14.98% | 1.34% | 9.36% | 11.60% |
| | ILA+PGD | 16/255 | 97.70% | 96.62% | 96.08% | 96.16% | 70.36% | 81.70% | 88.18% |
| | | 8/255 | 97.46% | 80.30% | 79.40% | 77.52% | 24.32% | 56.10% | 63.53% |
| | | 4/255 | 88.18% | 32.16% | 30.14% | 29.50% | 4.00% | 17.98% | 22.76% |
| | LinBP+PGD | 16/255 | **99.98%** | **98.38%** | **98.30%** | **97.86%** | **71.60%** | **85.98%** | **90.42%** |
| | | 8/255 | **99.92%** | **80.60%** | **80.38%** | **79.28%** | **26.04%** | **58.32%** | **64.92%** |
| | | 4/255 | **96.44%** | **33.04%** | **30.18%** | **29.94%** | **4.68%** | **19.20%** | **23.41%** |

Table 11: Success rates of transfer-based attacks on *CIFAR-10* using PGD with $\ell_\infty$ constraints. The source model is a *ResNet-18*. Average is obtained from models different from the source.

| Attack | Method | $\epsilon$ | VGG-19 (2015) | WRN (2016) | ResNeXt (2017) | DenseNet (2017) | PyramidNet[†] (2019) | GDAS (2019) | Average |
|---|---|---|---|---|---|---|---|---|---|
| Untargeted (ResNet-18) | PGD | 16/255 | 91.48% | 97.76% | 97.16% | 94.32% | 54.28% | 90.48% | 87.58% |
| | | 8/255 | 59.00% | 81.14% | 81.58% | 76.00% | 21.60% | 63.22% | 63.76% |
| | | 4/255 | 25.20% | 45.52% | 46.12% | 41.46% | 6.60% | 29.60% | 32.42% |
| | ILA+PGD | 16/255 | 94.20% | **98.08%** | **97.76%** | **95.88%** | 72.82% | 91.20% | 91.66% |
| | | 8/255 | 72.90% | 88.68% | 88.88% | 86.44% | 36.88% | 74.20% | 74.66% |
| | | 4/255 | 32.88% | 54.54% | 56.02% | 51.52% | 11.02% | 37.34% | 40.55% |
| | LinBP+PGD | 16/255 | **94.54%** | 96.54% | 95.42% | 93.98% | **78.48%** | **93.10%** | **92.01%** |
| | | 8/255 | **76.24%** | **89.76%** | **89.24%** | **87.28%** | **46.48%** | **78.62%** | **77.94%** |
| | | 4/255 | **37.74%** | **57.14%** | **62.66%** | **57.04%** | **15.18%** | **44.10%** | **45.64%** |
| Targeted (ResNet-18) | PGD | 16/255 | 62.32% | **84.34%** | 78.90% | 68.52% | 21.52% | 49.34% | 60.82% |
| | | 8/255 | 25.34% | 57.84% | 47.82% | 46.84% | 7.40% | 30.34% | 35.93% |
| | | 4/255 | 7.20% | 22.10% | 18.42% | 19.34% | 2.10% | 11.36% | 13.42% |
| | ILA+PGD | 16/255 | 51.48% | 66.36% | 54.32% | 50.80% | 29.98% | 27.06% | 46.67% |
| | | 8/255 | 34.06% | 60.20% | 49.98% | 46.92% | 12.82% | 31.98% | 39.33% |
| | | 4/255 | 9.98% | 25.88% | 21.04% | 21.12% | 3.60% | 13.64% | 15.88% |
| | LinBP+PGD | 16/255 | **75.06%** | 73.14% | 70.32% | 62.98% | **46.64%** | **63.66%** | **65.30%** |
| | | 8/255 | **47.22%** | **64.66%** | **62.92%** | **62.00%** | **21.38%** | **49.94%** | **51.35%** |
| | | 4/255 | **12.04%** | **27.50%** | **27.08%** | **28.22%** | **4.98%** | **18.24%** | **19.68%** |

Table 12: Success rates of transfer-based attacks on *ImageNet* using PGD with $\ell_\infty$ constraints. The source model is a *ResNet-50*. Average is obtained from models different from the source.

| Attack | Method | $\epsilon$ | ResNet* (2016) | Inception v3 (2016) | DenseNet (2017) | MobileNet v2 (2018) | PNASNet (2018) | SENet (2018) | Average |
|---|---|---|---|---|---|---|---|---|---|
| Untargeted (ResNet-50) | PGD | 16/255 | **100.00%** | 48.46% | 75.22% | 74.18% | 46.24% | 53.22% | 59.46% |
| | | 8/255 | **100.00%** | 21.10% | 43.38% | 44.92% | 17.64% | 22.56% | 29.92% |
| | | 4/255 | **100.00%** | 8.94% | 19.40% | 21.40% | 5.52% | 7.98% | 12.65% |
| | ILA+PGD | 16/255 | **100.00%** | 58.76% | 83.90% | 86.28% | 58.50% | 74.70% | 72.43% |
| | | 8/255 | **100.00%** | 28.94% | 54.86% | 58.02% | 27.12% | 38.96% | 41.58% |
| | | 4/255 | **100.00%** | 10.90% | 25.46% | 29.40% | 8.38% | 13.72% | 17.57% |
| | LinBP+PGD | 16/255 | **100.00%** | **83.66%** | **97.10%** | **96.80%** | **81.52%** | **89.66%** | **89.75%** |
| | | 8/255 | 99.98% | **42.12%** | **74.38%** | **74.22%** | **37.18%** | **51.72%** | **55.92%** |
| | | 4/255 | 99.92% | **13.74%** | **34.90%** | **36.02%** | **11.30%** | **17.34%** | **22.66%** |
| Targeted (ResNet-50) | PGD | 16/255 | **100.00%** | 0.04% | 0.44% | 0.28% | 0.16% | 0.24% | 0.23% |
| | | 8/255 | **100.00%** | 0.02% | 0.08% | 0.00% | 0.00% | 0.02% | 0.02% |
| | | 4/255 | 99.00% | 0.00% | 0.00% | 0.00% | **0.00%** | 0.02% | 0.00% |
| | ILA+PGD | 16/255 | 27.98% | 0.26% | 0.86% | 0.42% | 0.40% | 0.40% | 0.47% |
| | | 8/255 | 68.02% | 0.08% | 0.20% | 0.14% | 0.12% | 0.12% | 0.13% |
| | | 4/255 | 83.46% | **0.00%** | 0.08% | 0.02% | **0.00%** | 0.02% | 0.02% |
| | LinBP+PGD | 16/255 | 99.16% | **2.66%** | **13.42%** | **7.18%** | **4.84%** | **5.58%** | **6.74%** |
| | | 8/255 | 99.32% | **0.16%** | **2.08%** | **1.20%** | **0.48%** | **0.96%** | **0.98%** |
| | | 4/255 | 96.24% | **0.00%** | **0.14%** | **0.12%** | **0.00%** | **0.06%** | **0.06%** |

Table 13: Success rates of transfer-based attacks on *ImageNet* using PGD with $\ell_\infty$ constraints. The source model is an *Inception v3*. Average is obtained from models different from the source.

| Attack | Method | $\epsilon$ | ResNet (2016) | Inception v3* (2016) | DenseNet (2017) | MobileNet v2 (2018) | PNASNet (2018) | SENet (2018) | Average |
|---|---|---|---|---|---|---|---|---|---|
| Untargeted (Inception) | PGD | 16/255 | 47.46% | **100.00%** | 48.22% | 53.72% | 34.66% | 36.50% | 44.11% |
| | | 8/255 | 27.38% | **100.00%** | 27.26% | 33.56% | 15.70% | 17.00% | 24.18% |
| | | 4/255 | 13.64% | **99.96%** | 14.28% | 18.62% | 6.70% | 7.12% | 12.07% |
| | ILA+PGD | 16/255 | 84.94% | 99.80% | 80.26% | 89.28% | 65.60% | 74.70% | 78.96% |
| | | 8/255 | 55.42% | 99.68% | 50.06% | 64.12% | 33.82% | 42.16% | 49.12% |
| | | 4/255 | **25.96%** | 99.76% | **24.08%** | **34.44%** | 13.14% | **17.16%** | **22.96%** |
| | LinBP+PGD | 16/255 | **90.82%** | 99.98% | **89.24%** | **92.68%** | **78.02%** | **85.82%** | **87.32%** |
| | | 8/255 | **56.22%** | 99.76% | **55.46%** | **65.08%** | **34.78%** | **45.44%** | **51.40%** |
| | | 4/255 | 22.08% | 96.28% | 22.92% | 29.80% | 10.04% | 14.64% | 19.90% |
| Targeted (Inception) | PGD | 16/255 | 0.12% | **99.98%** | 0.20% | 0.08% | 0.16% | 0.12% | 0.14% |
| | | 8/255 | 0.02% | **99.96%** | 0.04% | 0.02% | 0.06% | 0.06% | 0.04% |
| | | 4/255 | 0.00% | **98.42%** | 0.02% | 0.00% | 0.00% | 0.00% | 0.00% |
| | ILA+PGD | 16/255 | **0.52%** | 2.02% | **0.64%** | 0.18% | **0.56%** | 0.26% | **0.43%** |
| | | 8/255 | **0.22%** | 14.54% | **0.30%** | **0.22%** | **0.28%** | 0.14% | **0.23%** |
| | | 4/255 | **0.06%** | 52.48% | **0.04%** | **0.06%** | **0.04%** | **0.06%** | **0.05%** |
| | LinBP+PGD | 16/255 | 0.34% | 4.26% | 0.44% | **0.26%** | 0.32% | **0.28%** | 0.33% |
| | | 8/255 | 0.10% | 6.08% | 0.12% | 0.14% | 0.06% | **0.20%** | 0.12% |
| | | 4/255 | 0.02% | 4.04% | 0.00% | 0.00% | 0.00% | 0.04% | 0.01% |