[Reviews · NeurIPS 2020]

Review 1

Summary and Contributions: This work introduces Linear Backpropagation (LinBP) for improving adversarial attack transferability, which changes the gradient w.r.t. data calculation in standard attacks such that the forward pass is unchanged but during backpropagation it ignores some of the nonlinear activation functions (i.e., if a ReLU had 0 activation in the forward pass, the gradient would not necessarily be zero'd out in the backward pass). The work is not proposed as a brand new attack algorithm, but is meant to be used with an existing attack like PGD and changes how the gradient w.r.t. the data is calculated. Further, it is motivated by a hypothesis of Goodfellow that adversarial attack transferability comes from linear nature of DNNs. The experimental results show that it can induce higher non-targeted attack transferability in both CIFAR10 and ImageNet models.

Strengths: - The method is reasonable, and the approach of LinBP is interesting to attack researchers. As correctly discussed in the RW, this is not the first work to leave the forward pass unchanged and manipulate the gradients in the backward pass (like SGM), but this method overcomes a weakness of SGM in that the source model does not have to contain residual connections (at the cost of some increased implementation complexity).

Weaknesses: - The transfer scenarios in Sec 3 are confusing, which in turn makes Figs 1&2 confusing. It seems like lines 108-110 state that VGG is always used as the whitebox source model and the WRN/RNXT/DN are always used as the victim blackbox target models. However, lines 128 - 130 contradict this talking about when WRN/RNXT/DN attack VGG? This critical detail is quite confusing. I would suggest to use transfer notation such as "VGG19 --> WRN" to clarify this both in the text and the figures. - Perhaps the key weakness for me is in the experiments. (1) The eps=0.1 from which the "99% average success rate" (from conclusion) is found is an unconventionally large epsilon in L_inf adversarial attacks. I would consider conforming to a more popular large epsilon such as eps=16/255 used in many papers so readers can roughly compare across works. I realize you also report eps=0.03 and 0.05 which is good, but these are not the focus of the discussion of results (e.g., paragraph line 228). (2) Testing with a variety of source models on both datasets should be strongly considered. Only using VGG19 for CIFAR10 and RN50 for ImageNet may lead to somewhat inconclusive results because the source model in transfer attacks matters. Does the method still work as well if you use a different source model? How does the choice of source model affect the transferability to each of the target models? I would be more interested to the answers of these questions for evaluations on ImageNet. Consider results of a contemporary paper: Table 3 of https://arxiv.org/pdf/2002.05990.pdf. This table shows 2 source models, 5 attacks (4 baselines + theirs), and 6 bbox target models, all of which is useful information. (3) Finally, the momentum iterative attack would be a much better baseline than IFGSM because it is designed as a simple (trivial) change to IFGSM that creates much more transferable adversarial examples essentially for free. It is used as a standard of measure in many many transfer attack papers but it is not used here at all. - There are no results discussing targeted attacks. Since this method reuses existing attack algorithms such as IFGSM, etc., creating targeted attacks is a trivial sign flip in the attack algo. For completeness of experiments, it would also be useful to report results of creating targeted attacks with the LinBP method. - Some aspects in the presentation quality of this paper are a weakness for a high quality publication (e.g. NeurIPS). For example, Figs 1&2 as discussed before, the tables with a "-" for the method, the "Dataset" columns in the tables are not informative, the management of Fig 3 and Table 2, a "*" appearing in Table 1 with no indication of meaning, etc.

Correctness: Yes, I believe that the LinBP method may in-fact boost transferability of adversarial examples. However, as discussed in the weaknesses, some additional experiments and use of a better baseline may make the results more convincing.

Clarity: Most parts of this paper reasonably written. However, see the last weakness point for further comments, as there are some confusing aspects of the notation in the figures and tables.

Relation to Prior Work: Yes, there is adequate discussion of related work.

Reproducibility: Yes

Additional Feedback: - In Fig 1a, what does it mean that the VGG19-LinS success rate is higher than the VGG19 success rate? Does this mean that effectively a blackbox attack (VGG19-LinS) is more successful than a whitebox attack (VGG19)? Or, is the VGG19 target model different from the source VGG19 model (i.e., a separately initialized VGG19)? If it does mean that a blackbox attack is better than a whitebox attack, why would this be the case? - In general, I think the notation used in Figs 1 & 2 could be clarified. What does WRN-LinS vs WRN mean? Is it "VGG19-LinS --> WRN" and "VGG19 --> WRN," respectively? May be helpful to use "VGG19-LinS --> WRN" notation or something similar so it is clear what the source and target models are. - Using the phrases "we try to" (lines: 79, 126, 138, ...) is rather unsatisfying. Rather, do you or don't you? Edit: and "probably" (lines: 138, 168) - Tables would read better if "I-FGSM" replaced "-". Also Linf eps = 0.1 is quite a large epsilon, would seem reasonable to use eps = 4/255, 8/255, 16/255, all of which are fairly common for attacking. - Table 2 seems most appropriate in the supplemental. - Is there a significant difference between using 100 attack iters and 10? ******************************************************************************************** EDIT after rebuttal ******************************************************************************************** Thank you for your thoughtful response to my concerns. As I mentioned, I think the method is interesting but the presentation and some key experimental choices require some serious re-working (in the main manuscript, not just in the supplemental). My suggestions are: - [p0] Improved source target transfer notation for clarity - [p0] Move attack results to 16/255, 8/255, 4/255 as to be more fairly compared across contemporary works. Also, remove any discussion of eps=0.1 as a main result, it is simply too high. - [p0] Include transfers with other source models for each dataset in the main tables. - [p1] Add some of the other baseline methods (e.g., momentum iterative) in the main tables - [p2] Include discussion of targeted attacks and/or transfers to robust models. This may be acceptable to go in the supplemental but it would be interesting in the main document if there is room. - Suggestions for saving space to accommodate above changes: It is unnecessary for figure 3 to take the whole column. Consider putting this in a wrapfigure with ~20% of the column width so there is more room for the above changes. Also Table 2 can be moved to the supplemental as it takes up a lot of room and provides almost no value.


Review 2

Summary and Contributions: The basic premise of the paper is that the transferability of adversarial attacks can be exploited through deliberate use of linear approximations. The approach, dubbed "LinBP", involves doing forward propagation as normal, but backpropagating "as if some nonlinear activations are not encountered in the forward pass". The core experimental claim is that this method "obviously outperforms current state-of-the-arts [sic] in crafting transferable adversarial examples on CIFAR-10 and ImageNet, leading to more effective attacks on a variety of DNN models". In the method section, the authors step back to the general "composition of matrix multiplication and elementwise nonlinearities" formulation of MLPs and their special case of convolutional networks, and point out that this provides a straightforward view on how one can make a "less nonlinear" network for a constant number of parameters, by simply omitting a choice of nonlinear activation functions. The argument (which I accept) is that this provides a good experimental framework for comparing the success of transfer attacks with respect to the inclusion or exclusion of nonlinear activations, all else equal. The initial experiment uses this framework to compute attacks on a CIFAR-10 VGG-19 which finishes with fully linearised layers, and transfer them to WRN, ResNeXt, and DenseNet. The partly-linearised VGG-19 is also fine-tuned for the original task before being used to compute transfer attacks (via FGSM and I-FGSM), and the relationship between the number of fine-tuning epochs and transferability is plotted. They note a quick initial improvement, followed by decline, w.r.t. the number of fine-tuning epochs. The key point here is that for a wide range of fine-tuning epochs (which are partly just to repair the hit to standard classification accuracy of the ad-hoc linearised model), transferability is noticeably increased compared to using the vanilla base model. They repeat the experiment with the same part-linearised architecture trained from scratch, noting that the results are similar for transferring across architectures, but different in that effectiveness is reduced w.r.t. the original f (i.e. that training from scratch appears to make the architecture as difficult to transfer attack as any other, when parameters are not initially shared). Using the above ("LinS") as an initial demonstration, they then present their actual proposal, linear backpropagation (LinBP), in which the fine-tuning is dropped and the scheme of omitting activation functions during backpropagation (alone) is used instead. They compare the proposal to the initial and optimally fine-tuned linearised networks in the previous demonstration, showing that LinBP is on par with the optimal tuning of LinS and far better than either standard transferred FGSM or I-FGSM attacks, or the initial (non-fine-tuned) LinSs. They also demonstrate a simple scheme for normalising the gradient contributions of the main and residual streams in the presence of skip connections. The main results in the paper concern the relative success in transferring I-FGSM attacks under different norm constraints from a CIFAR-10 VGG-19 and an ImageNet ResNet-50 onto a variety of other architectures, against the vanilla approach, TAP, ILA, and SGM. The variation in results with respect to the choice of base attack is handled in the supplementary material (DI^2-FGSM, PGD, ensemble attack), and is claimed to confirm the results given in the main paper (though I have not yet fully audited this portion of the claim by covering the supplementary material in detail). They further demonstrate plots of the performance of LinBP vs. ILA as a function of the location parameter in each, and finally, empirically demonstrate the success of combining LinBP with other methods (SGM and ILA), vs. LinBP or those methods on their own.

Strengths: Of course, all first-order attacks inherently, by definition, involve linear approximation of the networks under attack. However, the claim being made here is effectively that attacking a surrogate network whose local linear approximation would differ from that of the original network by "deactivating" ReLUs (i.e. allowing gradients to represent the effect on the output of moving in or out of the negative input half-line) leads to more transferable attacks. Considering that analytical gradients, in the context of optimisation, inherently suffer from the inability to represent nonlinear changes in the functions they describe and ergo do not accurately predict function values away from the point of approximation, and given the nature of ReLUs in the context of this fact (where such mispredictions can be severe when moving from the positive to negative half-line or vice versa), I consider the claim itself to be sensible enough to warrant investigation. (In the authors' words, "It can be regarded as feeding more gradient into some linear path than the nonlinear path that filters out the gradient of the negative input entries." Perhaps this wording could be tweaked, but this is the essential point.) Overall, I enjoyed this paper, and have added its results to my bank of knowledge on this topic. It is a good example of elegantly realising a well motivated intuition and providing ample evidence from well designed experiments in support of its hypothesis. I was especially impressed with the rigorous experimental methodology, particularly the plots of LinBP and ILA performance as functions of their respective location parameters, and the evaluation of method combinations. All results are meaningful and convincing. Are the topic and approach relevant to the NeurIPS community? Yes: this is a paper making an observation on the transferability of adversarial examples across models, and the relevance of that topic is not really controversial. I am not predicting an outpouring of wild enthusiasm from said community in response to this paper, but people who care about the details of network analysis would do well to read it. The presentation is, up to minor complaints (given in the "clarity" section), clear and well organised. I can easily recommend this paper for acceptance.

Weaknesses: I note no real weakness outside of the section on clarity of writing, and even this is not a particular weakness. (Again, its appeal will likely be limited to those who care about the non-sensational fine details of network analysis and adversarial vulnerability.)

Correctness: As above, they have passed all scrutiny to which I have been capable of subjecting them. The claims and method are straightforward, and I have found no holes in the experimental setup (which I would say has gone beyond the call of duty, if anything). The only caveat I can offer regarding my judgement (see also "Relation to prior work") is the fact that while possessing considerable expertise in the subfield of adversarial attacks/vulnerability, I do not specifically study "transfer attacks" in the sense considered here in particular detail. Thus, I cannot personally vouch that the experimental framework includes any and all valid "competitors", only that the framework itself is sound. Having said this, I find these results impressive, well motivated and substantiated, inherently plausible, and interesting in their own right.

Clarity: There are general points about English grammar and elegance of phrasing scattered throughout the paper too finely to be worth enumerating exhaustively. I would recommend that the paper be passed over by someone able to improve it in this regard. The paper is not poorly written, but it requires some polishing before publication. Having said that, see the below notes: 5: "improving... the linearity": odd wording/conceptualisation 41-42: "Many methods aim to maximize the prediction loss L(x + r; y) with a constraint on its Lp norm..." <- The constraint is on the Lp norm of the perturbation r, not the loss L. This wording needs to be fixed. 67: The word "literally" is jarring and not required. 121: "decent", not "descent" 137: "unlike", not "dislike" 212: "invoked", not "evoked" Also, the significance of the asterisk in "VGG-19*" in Figure 1 is not explained until much later, in the caption of Table 3. This statement needs to be made the first time the symbol appears. I spent more time than I should have parsing the first plot because of this. The captions can generally stand to be more detailed and explicit, as this will help readers who are less familiar with this material.

Relation to Prior Work: I would have listed ["Simple Black-box Adversarial Attacks", Guo et al., ICML 2019] as an example of a relatively query-efficient black-box attack (which does not represent an example of a transfer-based method), in the related work. The LinBP approach can be framed as an instantiation of BPDA ["Obfuscated Gradients Give a False Sense of Security: Circumventing Defenses to Adversarial Examples", Athalye et al., ICML 2018] in which the approximating network used in the backward pass is the "more linear" variant of the original. I would highly encourage the authors to make this connection explicit. As noted elsewhere regarding my own capacity as a reviewer, as I do not focus specifically on the problem of transfer attacks in my own research, I cannot personally verify the claim that the most relevant and recent state-of-the-art methods have been used in comparison. If there is any point of contention here from any other reviewers or meta-reviewers, it should be discussed.

Reproducibility: Yes

Additional Feedback: 18: The "blind spot" model of deep network vulnerability is questionable and arguably misleading, depending on how that statement is interpreted: see e.g. ["With Friends Like These, Who Needs Adversaries?", Jetley et al., NeurIPS 2018; "Adversarial Examples are not Bugs, they are Features", Ilyas et al., NeurIPS 2019]. I would discourage casually propagating this view (at least without any clarification). 18-19: "The undesirable phenomenon not only causes a great deal of concern when deploying DNN models in security-sensitive applications": Well, perhaps, but the general "security concern" statement, in the absence of any more specific example/argument, leaves this reviewer, who currently subscribes to the counterarguments to this point made in ["Motivating the Rules of the Game for Adversarial Example Research", Gilmer et al.], cold. I don't believe that this sort of boilerplate introduction to the problem of adversarial vulnerability is helpful or necessary. If there is a security case, state clearly what it is, else, don't make this statement. 27-28: "In this paper, we revisit the hypothesis of Goodfellow et al.’s [13] that the transferability (or say generalization ability) of adversarial examples comes from the “linear nature” of modern DNNs." I wouldn't completely agree that this was the argument in that paper. Those authors were, in my interpretation, arguing that the *existence/effectiveness* of adversarial examples could be explained by "excessive linearity" of networks, and accounted for how transferable perturbations are *between examples on the same network*. That is, they essentially foretold the existence of "universal adversarial perturbations" [Moosavi-Dezfooli et al., CVPR 2017]. They did note inter-network transferability, but noted that that phenomenon depended on different networks learning similar functions when trained to perform the same task, which is a somewhat different matter, even if there is a relationship. ---------------------------------------------------------------------- Update, following rebuttal and reviewer discussion ---------------------------------------------------------------------- After conferring with other reviewers, my score and overall opinion of the paper remain unchanged. Because my review was positive aside from specific, easily addressed points, the authors did not have to do much to address me and rightfully focused most of their efforts on R1's issues. I can concur with R1's requests for the following (copied verbatim from R1's updated review), the first of which I also made myself in my original review: " - [p0] Improved source target transfer notation for clarity - [p0] Move attack results to 16/255, 8/255, 4/255 as to be more fairly compared across contemporary works. Also, remove any discussion of eps=0.1 as a main result, it is simply too high. [R2: I am agnostic on the point about eps=0.1.] - [p0] Include transfers with other source models for each dataset in the main tables. " I believe these will increase the strength of what is already a strong paper, and so the authors should make an effort to include the results of their rebuttal in the updated text, which I'm sure they will (as they have already done the work and, unsurprisingly to me, realised favourable results). I would also ask that unless the authors have very strong reasons for overriding the specific objections that I had in the "additional feedback" section, to please take those points into account as well in the camera-ready version (which is more a matter of omission than inclusion). The same goes for the point about BPDA (though that is a matter of inclusion rather than omission). This is in addition to having the paper proofed for grammar and style issues.


Review 3

Summary and Contributions: In this paper, the authors propose to enhance the transferability of gradient-based black-box attacks by leveraging the linear nature of deep neural networks. The authors build upon and empirically study a hypothesis put forth by Goodfellow et al. [1], wherein the high transferability of black-box attacks is attributed to the linearity of networks. The paper proposes LinBP, where the forward pass is unchanged, but the loss is backpropagated linearly as though no non-linear activation function such as ReLU was encountered. Further, the authors demonstrate the improved transferability of different attacks on normally-trained victim models of diverse architectures. [1]: Goodfellow, I. J., Shlens, J., and Szegedy, C. Explaining and harnessing adversarial examples. In International Conference on Learning Representations (ICLR), 2015

Strengths: 1. First, the authors empirically study the hypothesis introduced in Goodfellow et al. [1], by proposing a method called linear substitution (LinS), where they fine-tune a normally trained network after removing the non-linear activations in the last few layers. 2. They show that both FGSM and IFGSM black-box attacks crafted from this linearised source model achieve a higher success rate in fooling other victim models that are trained with different architectures. 3. Motivated by these results, the authors propose a novel technique called linear backpropagation (LinBP), where the forward pass is unaltered, and the backward pass is performed as though non-linear activations were not encountered in the forward propagation. 4. The proposed method shows improved results when compared to previous works on both the CIFAR-10 and ImageNet datasets, with no significant additional overhead in computation. 5. The authors also demonstrate that their method can be combined with previous works to obtain a further boost in attack transfer. [1]: Goodfellow, I. J., Shlens, J., and Szegedy, C. Explaining and harnessing adversarial examples. In International Conference on Learning Representations (ICLR), 2015

Weaknesses: 1. For secured victim models, evaluations are performed only for a single ensemble adversarially trained Inception model; this is inadequate. 2. Could the authors clarify the exact training methodology used for ensemble adversarial training, such as the epsilon constraint, and the fixed source models used to generate the adversaries during training. 3. The authors have not included evaluations where the victim model is known to be robust, for example, white-box adversarially trained models using PGD [2], TRADES [3] or Feature Denoising [4]. [2]: Aleksander Madry, Aleksandar Makelov, Ludwig Schmidt, Tsipras Dimitris, and Adrian Vladu. Towards deep learning models resistant to adversarial attacks. In International Conference on Learning Representations (ICLR), 2018 [3]: Zhang, H., Yu, Y., Jiao, J., Xing, E. P., Ghaoui, L. E., and Jordan, M. I. Theoretically principled trade-off between robustness and accuracy. In International Conference on Machine Learning (ICML), 2019 [4]: Xie, C., Wu, Y., van der Maaten, L., Yuille, A., and He, K.Feature denoising for improving adversarial robustness, In IEEE Conference on Computer Vision and Pattern Recognition (CVPR), 2019

Correctness: Yes, the proposed method appears to be sound, and is adequately backed by valid empirical evaluations.

Clarity: Yes, the authors present their ideas in a clear, structured manner. Further, the paper is well written, and is easy to follow. Some minor corrections that can be incorporated have been indicated in the Additional Feedback section.

Relation to Prior Work: Yes, the paper does adequately compare and contrast their proposed method with previous published works. The paper expands upon certain ideas proposed in previous works, and presents distinguishing features in enough clarity.

Reproducibility: Yes

Additional Feedback: L167-169: “In later layers with a relatively large index i, Mi can be very sparse and removing it during backpropagation probably results in less gradient flow through the skip-connections” - Could the authors provide further clarity on this matter, particularly given that it motivates the need for the proposed re-normalisation term? Could the authors clarify how fine-tuning is performed for LinS in terms of the learning rate used etc? Could the authors include evaluations on multi-step adversarially trained models such as PGD [2] or TRADES [3] for CIFAR-10 and Feature Denoising [4] for ImageNet. It would be interesting to observe the attack success rates when the epsilon constraint assumed during training is followed/violated during test time. Some suggestions for minor errors in the paper: L93: “there is of yet few empirical evidence” → “there is of yet little empirical evidence” Eqn2: An extra closing parenthesis “)” can be removed L121: “also achieve descent transferability” - perhaps the authors meant “decent”? L160: “In the prequel of this paper” → “In the previous sections of this paper” The legend used in Figures-1, 2 could be presented with more clarity. For example, “WRN LinS” seems to suggest that the WRN model is linearised, rather than only the source VGG-19 model being modified. Table 6: If the authors primarily attribute the higher success rate seen for ILA to originate from the use of two-steps, could they include a two-step attack with LinBP for a fair comparison? For table captions, the authors could consider using “Success rates of transfer-based attacks” rather than “Performance of transfer-based attacks”. Could the authors include ablation studies for the re-normalisation factor alpha, such as fixing alpha to be 1 for all layers? [2]: Aleksander Madry, Aleksandar Makelov, Ludwig Schmidt, Tsipras Dimitris, and Adrian Vladu. Towards deep learning models resistant to adversarial attacks. In International Conference on Learning Representations (ICLR), 2018 [3]: Zhang, H., Yu, Y., Jiao, J., Xing, E. P., Ghaoui, L. E., and Jordan, M. I. Theoretically principled trade-off between robustness and accuracy. In International Conference on Machine Learning (ICML), 2019 [4]: Xie, C., Wu, Y., van der Maaten, L., Yuille, A., and He, K.Feature denoising for improving adversarial robustness, In IEEE Conference on Computer Vision and Pattern Recognition (CVPR), 2019 Post Rebuttal: The authors seem to have adequately addressed the concerns of all four reviewers. The additional results with a robust target model and different source models further highlight the superior performance of the proposed attack compared to previous works.


Review 4

Summary and Contributions: This paper proposes a linear backpropagation (LinBP) technic which backpropagates loss as if there is no non-linear activation between the weight matrix. The proposed method is simple but shows its effectiveness on the transferability of adversarial examples.

Strengths: 1. The key idea of linear backpropagation seems novel and interesting. 2. Empirically validated the effectiveness of linear backpropagation technic.

Weaknesses: The linearity hypothesis is validated in an empirical manner, so more solid experiments will make this paper more convincing. 1. Can other network architectures (ResNet [1] or DenseNet [2] etc.) also transfer adversarial examples well as in VGG-Net with LinBP? In other words, please show the generalization ability of LinBP method. 2. In LinS and LinBP, why only the non-linearity of the two last block of VGG-Net was removed? What if more than two layers are selected? or, why the first blocks cannot be selected? If there are some insights, theoritical reason, or empirical results, please provide them. [1] He et al., "Deep Residual Learning for Image Recognition" [2] Huang et al., "Densely Connected Convolutional Networks"

Correctness: The authors empirically shows the effectiveness of the method.

Clarity: Yes.

Relation to Prior Work: I guess so.

Reproducibility: Yes

Additional Feedback: The rating would be changed to accept if the authors resolve all the questions in the weakness section. * Post-rebuttal comment I would like to keep my original rating as the author's response did address my concerns well.

[Author Response · NeurIPS 2020]

We would like to thank all the reviewers for providing valuable feedback. Below are our responses to the comments.

**Reviewer#1:** 1) To the comment "the transfer scenarios in Sec 3 are confusing", we would like to explain that VGG-19 was indeed always used as the source model in Sec 3 in our paper. On lines 128–130, we meant to say that if we had trained the LinS model from scratch, the success rate of using it to attack VGG-19 (as shown in the grey curve in Figure 2) would have been much lower than fine-tuning (as shown in the grey curve in Figure 1). We will change the legends in Figure 1 and 2 to "VGG-19 → WRN", "VGG-19-LinS → WRN", etc. as suggested.

2) We followed the suggestion of performing experiments with $\epsilon$=16/255, 8/255, and 4/255. On ImageNet, our LinBP achieved an average success rate of 84.77%, 50.56%, and 20.89%, respectively, showing that it still outperformed the other methods (e.g., ILA: 72.34%, 42.05%, and 17.60%) remarkably. In addition, when combined with ILA and SGM, our method further gained an average success rate of **90.20%** under $\epsilon$=16/255. As has been recognized by the reviewer, we also reported results under $\epsilon$=0.05 and 0.03 in our paper, and we will discuss the results further in these settings that lead to more imperceptible perturbations in the final version of the paper.

3) We followed the suggestion of testing with the momentum iterative FGSM (MI-FGSM) attack on both CIFAR-10 and ImageNet, and the superiority of our LinBP still held as with I-FGSM. Specifically, our LinBP achieved an average success rate of 87.50% on ImageNet while the second best method (i.e., ILA) achieved 71.21% in the untargeted setting under $\epsilon$=16/255 (see Table 1b). We also reported the results using other baseline attacks (i.e., DI$^2$-FGSM, PGD, and an ensemble attack) in the supplementary material of the paper, which further demonstrate the effectiveness of our method.

4) We considered two other source models as suggested: ResNet-18 (on CIFAR-10) and Inception v3 (on ImageNet). With these two models, our method outperformed its competitors similarly under the constraint of $\epsilon$=16/255, 8/255, and 4/255. See Table 1a and Table 1c for the detailed results.

5) We followed the suggestion of discussing targeted attacks. Table 1 shows that the superiority of our method holds on both CIFAR-10 and ImageNet in the targeted setting as well. Due to the space limit, we only compared our method with the baseline attack and the second best method in the table.

Table 1: More results of the transfer-based attacks on CIFAR-10 and ImageNet, using MI-FGSM as the baseline attack.

| Source | Method | $\epsilon$ | Untargeted | Targeted | Source | Method | $\epsilon$ | Untargeted | Targeted | Source | Method | $\epsilon$ | Untargeted | Targeted |
|---|---|---|---|---|---|---|---|---|---|---|---|---|---|---|
| ResNet-18 (CIFAR-10) | MI-FGSM | 16/255 | 84.35% | 40.87% | ResNet-50 (ImageNet) | MI-FGSM | 16/255 | 58.67% | 0.17% | Inception v3 (ImageNet) | MI-FGSM | 16/255 | 48.44% | 0.15% |
| | | 8/255 | 62.68% | 28.84% | | | 8/255 | 34.51% | 0.06% | | | 8/255 | 31.16% | 0.06% |
| | | 4/255 | 34.00% | 12.68% | | | 4/255 | 16.94% | 0.01% | | | 4/255 | 17.00% | 0.01% |
| | ILA | 16/255 | 90.26% | 39.19% | | ILA | 16/255 | 71.21% | 0.34% | | ILA | 16/255 | 75.04% | 0.25% |
| | | 8/255 | 73.75% | 33.69% | | | 8/255 | 40.84% | 0.07% | | | 8/255 | 46.78% | 0.14% |
| | | 4/255 | 38.90% | 14.49% | | | 4/255 | 17.86% | 0.02% | | | 4/255 | 21.95% | 0.02% |
| | LinBP (ours) | 16/255 | **94.03%** | **71.66%** | | LinBP (ours) | 16/255 | **87.50%** | **5.01%** | | LinBP (ours) | 16/255 | **81.07%** | **0.35%** |
| | | 8/255 | **81.11%** | **57.24%** | | | 8/255 | **55.87%** | **0.93%** | | | 8/255 | **48.26%** | **0.17%** |
| | | 4/255 | **47.32%** | **22.25%** | | | 4/255 | **25.16%** | **0.06%** | | | 4/255 | **22.56%** | **0.11%** |

(a) CIFAR-10: ResNet-18 → victims     (b) ImageNet: ResNet-50 → victims     (c) ImageNet: Inception v3 → victims

6) We ran attacks for 100 iterations to ensure that all the methods achieved their best performance. The success rate of the methods decreased 5%-20% if we ran only 10 iterations. Indeed, sometimes our LinBP achieved slightly higher attack success rates in attacking the source models than those of I-FGSM, similar to an observation made in the ILA paper. This is likely because analytical gradients cannot represent nonlinear functional changes of $f$ (caused by each perturbation step, which is as large as 1/255), as commented by Reviewer#2.

**Reviewer#2:** We will discuss the mentioned related work. Thanks for the reference.

**Reviewer#3:** 1) We followed the suggestion of attacking a black-box robust ResNet (https://bit.ly/2C9FJVM) guarded by PGD adversarial training. The experiment shows similarity superiority of attack using LinBP (victim error rate: 48.60%, 39.92%, and 37.10% with $\epsilon$=0.1, 0.05, 0.03) to ILA (42.30%, 37.82%, and 36.60%). The model in Sec 5.2 guarded by ensemble adversarial training was obtained on GitHub (https://bit.ly/2XKfrkz), provided by Kurakin et al. 2) Without re-normalization, the performance of our method degraded to 81.36% (from 96.89%), under $\epsilon$=0.1. The norm of gradient became much larger in the main stream of the residual network with $W_iW_{i+1}$ being calculated instead of $W_iM_iW_{i+1}$, so that the gradient flowing through the main stream dominated, which is undesirable according to SGM. 3) We followed the policy of fine-tuning in a PyTorch tutorial, and more details will be included in an updated version of the paper. 4) With a two step policy, our LinBP indeed achieved higher success rates (90.49%, 74.44%, and 52.95%) than those of ILA in Table 6.

**Reviewer#4:** 1) We compared different methods on VGG-19/ResNet-18 on CIFAR-10 and ResNet-50/Inception v3 on ImageNet (see Table 1 in this response). It can be seen that the superiority of our method holds on all these concerned architectures. 2) Our method was invoked at the same positions as for ILA for fairness. Our paper discussed how the performance of our method and LinBP varied with the choice of positions in Figure 4.

[Meta-Review · NeurIPS 2020]

The paper proposes to improve the transferability of adversarial examples by using linear approximations of DNNs. Although there are still some requests from reviewers to improve the presentation (please take their advice into account, it will increase the impact of your paper), there is a consensus that the experimental evaluations of the method are complete and sufficiently robust to convince of the soundness of the methods and its working hypotheses.